# Negative Pressure Wound Therapy for the Treatment of Fournier’s Gangrene: A Rare Case with Rectal Fistula and Systematic Review of the Literature

**DOI:** 10.3390/jpm12101695

**Published:** 2022-10-11

**Authors:** Michele Altomare, Laura Benuzzi, Mattia Molteni, Francesco Virdis, Andrea Spota, Stefano Piero Bernardo Cioffi, Elisa Reitano, Federica Renzi, Osvaldo Chiara, Giovanni Sesana, Stefania Cimbanassi

**Affiliations:** 1Department of Surgical Sciences, Sapienza University of Rome, Piazzale Aldo Moro 5, 00185 Rome, Italy; 2General Surgery and Trauma Team, ASST Niguarda, Milano, Piazza Ospedale Maggiore 3, 20162 Milan, Italy; 3General Surgery Residency Program, University of Milan, Via Festa del Perdono 7, 20122 Milan, Italy; 4Department of Surgery, University Vita-Salute, San Raffaele Scientific Institute (HSR), 20132 Milan, Italy; 5Ospedale Maggiore Della Carità di Novara, 28100 Novara, Italy; 6Department of Medical-Surgical Physiopathology and Transplantation, University of Milan, Festa del Perdono 7, 20122 Milan, Italy; 7Tissue Bank and Tissue Therapy, ASST Niguarda, Milano, Piazza Ospedale Maggiore 3, 20162 Milan, Italy

**Keywords:** case report, Negative Pressure Wound Therapy (NPWT), Fournier’s gangrene, rectal fistula, hyperbaric oxygen therapy, surgical technique

## Abstract

Fournier’s gangrene (FG) is a Necrotizing Soft Tissue Infection (NSTI) of the perineal region characterized by high morbidity and mortality even if appropriately treated. The main treatment strategies are surgical debridement, broad-spectrum antibiotics, hyperbaric oxygen therapy, NPWT (Negative Pressure Wound Therapy), and plastic surgery reconstruction. We present the case of a 50-year-old woman with an NSTI of the abdomen, pelvis, and perineal region associated with a rectal fistula referred to our department. After surgical debridement and a diverting blow-out colostomy, an NPWT system composed of two sponges connected by a bridge through a rectal fistula was performed. Our target was to obtain healing in a lateral-to-medial direction instead of depth-to-surface to prevent the enlargement of the rectal fistula, promoting granulation tissue growth towards the rectum. This eso-endo-NPWT technique allowed for the primary suture of the perineal wounds bilaterally, simultaneously treating the rectal fistula and the perineum lesions. A systematic review of the literature underlines the spreading of NPWT and its effects.

## 1. Introduction

Necrotizing soft tissue infections (NSTIs) are life-threatening complications of the most common skin and soft tissue infections (SSTIs) with a high morbidity and mortality rate. Limited tissue necrosis may lead to septic shock followed by multiple organ failure. In the last decades, the mortality rate has slowly decreased due to widespread awareness of the disease and improved care, but the mortality range is still crucial, from 20% to 80% worldwide [1]. NSTIs may affect any part of the body. The genital, perianal, and perineal region involvement is known as Fournier’s gangrene (FG), with a reported mortality rate of 20–50% [1]. In FG, the infection is more frequently polymicrobial due to aerobic and anaerobic bacteria. Fournier’s gangrene arises from anorectal or genitourinary tract infections which may spread to the thighs, the anterior abdominal wall, and the retroperitoneum. The main risk factors described in the literature are gender male predominance, diabetes mellitus, immunosuppression and alcoholic liver disease. The FG incidence is higher for male patients with a mean age of 60 [2] even if a higher mortality risk is described in females [3,4]. The diagnosis is mainly clinical according to signs and symptoms [5]. Laboratory findings and imaging may be helpful, and many scores have been described. The most commonly used are the Laboratory Risk Indicator for Necrotizing Fasciitis (LRINEC) score and the computed tomography-based scoring system for soft-tissue infections [6,7,8]. The Fournier’s Gangrene Severity Index (FGSI) Score was developed to precisely predict the prognosis in patients with FG [9]. The source control of NSTIs requires three combined aspects: early surgical incision with debridement, broad-spectrum antimicrobial therapy, and intensive management. A complete debridement is rarely reached with a single surgical exploration. Tissue samples should be taken for cultures from the first debridement. Re-explorations are recommended every 24–48 h. When the infection involves the perineal region, diverting colostomy and urostomy should be considered. According to the World Society of Emergency Surgery (WSES) and the Surgical Infection Society Europe (SIS-E), negative pressure wound therapy (NPWT) is recommended after the complete removal of necrotic tissue [1]. Clinical evidence of NPWT’s additional benefit compared with conventional wound treatment has not been proven yet [10,11]. Post-surgery hyperbaric oxygen (HBO) therapy uses 100% pure oxygen at a pressure of 2–3 absolute atmospheres, which results in enhanced oxygenation of blood and tissues, improving wound healing. HBO can be used, if available, even though there is insufficient evidence supporting its benefits. 

This study aims to describe a rare case of FG with a rectal fistula referred to our hospital. The patient was successfully treated with combined eso-endo lumen NPWT. A systematic review of the literature underlines the spreading use of NPWT and its effects.

## 2. Materials and Methods

The article has been conducted following the Preferred Reporting Items for Systematic reviews and Meta-Analyses (PRISMA) statement. We followed the PRISMA update published in 2020 [12].

### 2.1. Eligibility Criteria

We limited the inclusion criteria to all the articles published from January 2009 to December 2021 describing NSTIs and FGs treated with NPWT. Only English-language publications with full text available were included. All of the reviews of the literature and the articles describing all of the patients with NSTIs and FGs treated without NPWT were excluded.

### 2.2. Information Sources and Search Strategy

We conducted the research in MEDLINE (OVID), EMBASE, and Central Cochrane Controlled Trials Register (CENTRAL). The following keywords were used to search in titles and abstracts: “Necrotizing Soft Tissue Infection,”; “Fournier’s Gangrene,”; “Negative Pressure Wound Therapy,” or “Topical Negative Pressure Therapy,” or “Negative Pressure Dressing,” or “Vacuum-Assisted Closure.” Appendix A shows the details of the research process. No other filters were used. The sources were last consulted on the 15 August 2022.

### 2.3. Selection Process

Titles and abstracts selection was performed independently by two reviewers with the abovementioned criteria. The full-text articles were reviewed for the final inclusion. When more than one article was reported by the same institution and authors, we selected the one with the most extensive series and the most recent. A PRISMA flow-chart is reported in Figure 1. 

### 2.4. Data Collection Process and Data Items

The included items were the following: the type of the study (case report, case series, cohort study); the year of publication; the number of patients included; the site of origin of NSTIs (anorectal, urogenital, or undefined); the microbiological etiology (monomicrobial or polymicrobial); the primary diameter of the lesion; the antibiotic therapy administered; the anatomical spread of the infection (limited to the perineum or abdominoperineal); the clinical scores used (Fournier’s Gangrene Score Indexand LRINEC score, and Neutrophil to Lymphocyte Ratio (NLR)); treatment modality (surgical debridement, hyperbaric oxygen therapy, NPWT, enterostomy, etc.); the length of hospital stay; the time from the first debridement to the wound closure.

### 2.5. Risk of Bias Assessment

All the articles included were either case reports or case series limited by their retrospective nature. We conducted our research on three databases, considering the rarity of the pathology. Since NSTIs are rare complications, only small samples of patients were enrolled in each publication. We assessed the risk of bias in the included studies through the STrengthening the Reporting of Observational Studies in Epidemiology (STROBE) statement [13]. A meta-analysis was not possible due to the clinical and methodological heterogeneity of the studies. 

## 3. Results

### Characteristics of the Included Studies

We found 58 eligible articles from the databases’ automatic search and 14 relevant articles from the hand-search. After the selection process, 28 articles were included (21 case reports or case series and 7 cohort studies). Data extracted from each study are summed up in Appendix B.

We present the main characteristics of the included studies:

Ozturk et al. [14] (2009) presented a retrospective analysis of 10 patients suffering from FG. After surgical debridement and broad-spectrum antibiotics, half of the patients were treated with the traditional dressing and half with the NPWT system. Polymicrobial etiology was found in 80% of cases of each group. Patients in the NPWT group reported less pain and less need for analgesics had greater mobility, missed fewer meals, and needed more occasional dressing changes than patients in the traditional group. Even though no difference was noticed in the length of stay and in time from surgical debridement to wound healing, the NPWT group needed fewer dressing changes, better pain relief, and general patient satisfaction. Cuccia et al. [15] (2009) performed a retrospective review of six patients with FG. The severity of infection was evaluated with the FGSI score. After surgical debridement, the authors used hyperbaric oxygen therapy and NPWT to prepare the wound area for the reconstructive phase. According to this study, NPWT reduced the number of surgical debridements and resulted in a simple solution for resurfacing significant scrotal defects with a cosmetically and functionally good result. Negative pressure is a time-saving device that gives reliable and repeatable results, reducing admission length and patient discomfort. Tucci et al. [16] (2009) described two cases of ischio-rectal and perineal NSTI in female patients successfully treated with the NPWT system. The authors promote NPWT, which can be changed every 48–72 and is less painful and more comfortable for patients than traditional dressings. Czymek et al. [4] (2010) designed a cohort study including 38 patients with FG. The authors describe two periods from January 1996 to January 2002 and from February 2002 to February 2008 showing how the use of NPWT had increased (33.3% versus 60.9%). The authors consider NPWT really effective, according to their clinical experience. Wagner et al. [17] (2011) reviewed a case series of 41 patients with FG. Each patient was treated with surgical debridement, broad-spectrum antibiotics, NPWT, and HBO. The severity of FG was evaluated using the FGSI score. Each patient recovered completely after one month even though the median FGSI score was low, reflecting the low severity of the infection in this series of patients. Pour et al. [18] (2011) described a single case of NSTI involving the perineal area and the right lower abdominal quadrant. The treatment consisted of surgical debridement and broad-spectrum antibiotics. In this case, NPWT was used to prepare the wound for the reconstructive phase for twelve days. Total wound closure was completed in two months. Zagli et al. [19] (2011) presented two cases of FG successfully treated with combination therapy of surgical debridement, broad-spectrum antibiotics, diverting stoma, NPWT, and HBO. They underlined the synergistic effect of these strategies. Jones et al. [20] (2012) presented 3 cases of NSTIs of urogenital origin. A polymicrobial etiology was diagnosed, and the patients were treated with surgical debridement, antimicrobial therapy, and NPWT. Reconstructive surgery and stoma diversion, either fecal or urinary, were not necessary. They reported a median length of stay of 5 days and obtained the wound closure in 190 days. The authors concluded that the application of NPWT after surgical debridement had the potential to remove tissue exudate, reduce the local edema, enhance neovascularization, and improve the natural self-healing ability. In this article, the NPWT promoted the patient’s comfort, reducing the need for more frequent dressing changes. Pastore et al. [21] (2013) described a multi-step approach to managing FG with surgical debridement, HBO therapy, and NPWT. NPWT helped to control the infection and the induction of the new granulation tissue. Moreover, reconstructive surgery was not necessary. After 34 days of NPWT, the patient was discharged with the surgical wound almost completely healed. Agostini et al. [22] (2014) presented a single case of FG that was treated with surgical debridement, broad-spectrum antibiotics, diverting colostomy, and suprapubic cystostomy. After this stage, NPWT, dermal regeneration template, and split-thickness skin graft were used. The patient also received daily HBO therapy. Etiology was polymicrobial. After 21 days of NPWT, the patient was ready for reconstruction. The complete healing required 58 days. Ludolph et al. [23] (2014) described three cases of FG of the perineum requiring penile skin removal with the need for reconstruction. Concerning the replacement of penile skin with split-thickness skin grafts, the authors suggested the creation of an additional neo-tissue layer using biomatrices such as collagen templates: to advance the incorporation of an acellular dermal matrix, the application of NPWT has proved to be a reliable and safe tool to stabilize and secure the grafts during the initial healing phase. Lee et al. [24] (2014) presented a retrospective analysis of 8 patients suffering from NSTI treated with a combination of NPWT and dermatotraction in a shoelace manner using elastic vessel loops after surgical debridement. Extended NPWT-assisted dermatotraction advances scarred, stiff fasciotomy wound margins synergistically in NSTI and allows direct wound closure without complications. This method is suitable for large open wound closure in patients with poor general conditions that cannot undergo complex reconstructive procedures. Ye et al. [25] (2014) described a double-phase use of NPWT. After surgical debridement of a perineal NSTI, NPWT was applied to prepare the wound bed; then, a split-thickness skin graft was prepared to cover the wound, and NPWT was used to improve the chances of graft acceptance. Oymaci et al. [26] (2014) reported a case series of 16 patients with FG. FGSI score was used to grade the severity of infection. In those cases where NPWT was used: ten out of sixteen patients underwent primary wound closure, two underwent secondary closure, and one was still on treatment when the study was published. In this study, the number of consecutive dressings was decreased, and skin defects were primarily closed at earlier periods. Oguz A et al. [27] (2015) presented a case series of 43 patients with FG. In this paper, the authors confirmed the importance of the FGSI score in determining the severity of infection. NPWT was applied to eight patients, demonstrating its advantages in patient satisfaction and compliance with the treatment strategy according to other studies already discussed. Ozkan et al. [28] (2016) presented a retrospective analysis of twelve patients with FG’s diagnosis. The etiology was anorectal in eight patients, urogenital in three, and unclear in one. Polymicrobial infection was isolated in 50% of the patients. Six patients needed fecal diversion. After the debridement, NPWT was used in four patients, while the conventional dressing was used for the others. The mean hospital stay was 18 days in the NPWT group and 20 days in the traditional group of dressing. This study underlined the positive effects of NPWT to help wound healing physiologically with less frequent changes. Emre et al. [29] (2016) reported a case of FG of the perineum in a cachectic patient that underwent surgical debridement, broad-spectrum antibiotic regimen, colostomy, and NPWT application for 45 days. After this period, the wound area was ready to be reconstructed by partial-thickness split graft and discharged home after 30 days. Yanaral et al. [30] (2017) evaluated 54 patients with FG. After surgical debridement and the administration of antibiotics, patients were divided into two groups (conventional dressing and NPWT). This study did not demonstrate that NPWT leads to a better outcome to patients but underlined some advantages such as fewer dressing changes, less pain, and greater mobility. 

Misiakos et al. [31] (2017) retrospectively reviewed a case series of 62 patients affected by NSTI. All patients underwent surgical debridement and broad-spectrum antibiotic therapy. The use of the NPWT system to accelerate wound healing was reported in four cases only. The authors did not notice a reduction in length of stay. The authors suggested that by combining the higher cost of NPWT therapy with conventional gauze therapy, NPWT should be used only in wounds with large surfaces and/or in patients with several comorbidities. LRINEC was used for the diagnosis of NSTI and as a severity score. Hong et al. [32] (2017) retrospectively reviewed twenty FG cases, describing each case’s treatment strategy. NPWT was used in two patients with extensive wound surfaces to reduce it, avoiding the need for reconstructive surgery. Yucel et al. [33] (2017) performed a retrospective analysis of 25 cases of FG treated with surgical debridement, broad-spectrum antibiotics, and NPWT in the most complicated cases. NPWT provided better-wound care and better patient satisfaction because of fewer dressing changes needed but was associated with longer length of stay and more debridements; this was explained by the greater extension and degree of infection in cases where NPWT was applied. Chang et al. [34] (2018) reviewed a case series of 13 patients affected by FG treated with NPWT after surgical debridement without reconstructive procedures. Tian et al. [35] (2018) described a case of FG of the perineum, successfully treated. The NPWT was applied to a split-thickness skin graft to improve graft survival and cover the anus to prevent fecal contamination avoiding a fecal diversion system. Syllaios et al. [36] (2020) presented a single case of NSTI of the scrotum and the perineal area. Surgical debridement, broad-spectrum antibiotics, and a loop colostomy were performed. On the third postoperative day, NPWT was applied. The wound was closed on the 13th postoperative day. NPWT leads to fewer dressing changes, less pain, fewer skipped meals, greater mobility, reduced hands-on treatment time for the clinician, and a shorter hospital stay compared to the conventional method. Zhang et al. [37] (2020) reported a case series of 12 patients with FG with an average age of 60 years old. The NPWT system was applied in ten cases to facilitate wound healing. A disadvantage of NPWT applied to the perineum is that fecal contamination leads to leaks in the vacuum. The authors suggest the use of the LRINEC score to help with the diagnosis of infection. Kostovski et al. [38] (2021) reported a single case of FG and the use of NPWT as a treatment strategy. Gul et al. [39] (2021) revised 22 cases of FG treated with surgical debridement. NPWT was used in twelve patients. Even though they recognized the utility of vacuum therapy in wound healing, it did not lead to a statistically significant benefit for mortality and morbidity. Iacovelli et al. [40] (2021) presented a multi-center cohort study with 92 patients affected by local or disseminated FG. After surgical debridement, those patients were divided into two groups according to wound healing management: conventional dressings versus NPWT. The length of stay was longer in the NPWT group (both local and disseminated FG) than in traditional dressing. The authors demonstrated that NPWT leads to faster wound healing at ten weeks in disseminated gangrene (with no difference in local gangrene) and higher overall survival at 90-days.

## 4. Case Presentation

We describe the case of a 50-year-old woman with a NSTI of the perineal region extended to the abdomen and the inguinal canal. 

She was referred to our department from a tertiary center where the patient went for abdominal and perineal pain associated with fever (38.5 °C). The medical history was characterized only by a cesarean section thirty years before. The patient was hemodynamically stable. A prompt clinical diagnosis of Fournier’s Gangrene was made and confirmed by CT. An early surgical debridement with broad-spectrum antimicrobial therapy was performed. The infection involved skin and soft tissue of the perineal region extended to the right inguinal canal and the abdomen on the right and left flank (Figure 2). A rectal fistula was diagnosed and a seton was placed. Tissue samples were taken for cultures and resulted positive for *Enterococcus raffinosus*. A specific antibiotic regimen started with Meropenem, Tigecycline and Fluconazole. The patient underwent re-explorations and debridement every 24–48 h. The patient was recovered in the Intensive Care Unit and a rectal tube was used the first few days.

The patient was referred to our department after eight days. In our hospital, a new CT scan was performed and showed collections with hydro-aerial levels closed to the posterior wall of the rectum. We continued with the debridement every 24–48 h, performing a transverse blow-out colostomy to promote the daily medications. The fistula seton was removed and a rectal exploration showed a defect of almost 2 cm of the posterior rectal wall at 5–6 o’clock. Other tissue cultures were taken and were positive for *Enterococcus raffinosus* and *Proteus mirabilis*, but no changes in antibiotic therapy were necessary. After 17 days from the first operation, no more collections were detected, the tissues were cleaned and the NPWT was placed. In the beginning, the NPWT was composed of two abdominal dressings on the left and the right side connected to the right perineal cavity with one sponge into the right inguinal canal. A third dressing was used to cover the left perineal wound. The pressure was maintained constant at −125 mmHg. Re-explorations were made every 48–72 h for the first two weeks. After four days with NPWT (21 days from the first operation), the right and left abdominal wounds were closed. Then a sponge was created to connect the right and went through perineal cavities through the rectal fistula. This eso-endo NPWT solution allowed for the growth of the granulation tissue in the left perineal cavity, preventing the endoluminal hole’s enlargement. The 3-step placement of the eso-endo-NPWT is shown and explained in Figure 3. We continued the same antibiotic therapy for four weeks. In the meantime, the patient was also treated with hyperbaric therapy sessions from the first day in our hospital to the last day of NPWT. NPWT was applied for thirty-two days, and the perineal wounds were closed bilaterally. A rectoscopy was performed, and no rectal defects were detached. Fifty-two days after the diagnosis of FG and the first surgery, the patient was discharged with the indication to perform an abdominal Magnetic Resonance Imaging (MRI) in one month. It showed a complicated trans sphincteric fistula and the absence of collections. The fistula was treated with a seton (Figure 4) and is now completely healed. We are planning to close the colostomy soon.

## 5. Discussion

Necrotizing soft tissue infections are rare and life-threatening bacterial infections with diagnostic and therapeutic challenges and high mortality and morbidity rates. The primary keys of NSTI management are well-defined and are described in all of the included studies as well in our case report. The basics are prompt surgery debridement, antibiotic therapy, and supportive care. Despite the well-known basics of the management, the survival of patients with NSTIs is highly variable depending on the disease’s characteristics and factors over the treatment. Fournier’s gangrene is a type of NSTI affecting the genital, perianal, and perineal regions. Many studies classify the FG extension as local if it is confined to the pubic area or disseminated if it extends to another body region such as the abdomen, thighs, etc. [14,40]. In our case, the patient developed a disseminated FG of the perineal region extended to the right inguinal canal and the abdomen on the right and left flank. Age over 50 years is considered a predisposing factor for FG. Most of the included studies found an average age of 60 years old [37,40]. Moreover, FG tends to occur in patients with immune disorders, diabetics, overweight and alcoholic liver disease, who have more significant difficulties in wound healing. According to the literature, a polymicrobial etiology has been recognized in most of the included studies such as in our case. The clinical diagnosis is usually supported with the LRINEC score, while the FGSI score is not frequently applied. The progress in surgical management and the development of new therapies promoting wound healing are slightly increasing the overall survival. Negative Pressure Wound Therapy is a wound dressing attached to a vacuum suction machine and is used more and more frequently. The debridement should be almost concluded when NPWT is applied. Yanaral et al. showed how NPWT effectively offers fewer dressing changes, less pain, and greater mobility [30]. Jones et al. described how NPWT had the potential to remove tissue exudate, reduce the local edema, enhance neovascularization, improve the natural self-healing ability, and promote the patient’s comfort, reducing the need for frequent dressing changes [20]. The association of NPWT and complex tissue reconstructions such as skin grafts or dermal matrices can be safe and reliable [22,23,35]. On the other hand, the early use of NPWT can reduce the need for reconstructive surgery [21,32]. Iacovelli et al. observed a significant difference in wound closure rates in patients with disseminated FG treated with NPWT. On the contrary, local FG patients did not show the same advantages [40]. Misiakos et al. suggested that NPWT should be used only in wounds with large surfaces and in patients with several comorbidities. The expensive costs of this suction dressing could be considered, but it is challenging to compare prices in different countries and hospitals [31]. Patients undergoing VAC had significantly longer durations of hospital stay and a higher mean number of debridements performed [33]. In our case report, NPWT was applied for faster wound healing. In consideration of the perianal fistula, finding an eso-endo NPWT solution turned out to be effective. Despite the lack of evidence, synergistic effects with HBO should not be underestimated. All of the described results of NPWT in FG are retrospective. Statistically significant results are rare and characterized by small cohorts leading to low-level evidence papers. There are several limitations in this study. First of all, the heterogeneity and the low quality of evidence of all of the included studies do not lead to reliable conclusions. On the other hand, this paper’s principal aim was to underline the multiplicity of treatments used for FG without few strong recommendations. The lack of clear and strong guidelines in this field is relevant and needs to be overcome. Moreover, a case report is surely not the best option to prove the efficacy of one treatment. Otherwise, it seems to us that the presentation of this rare case of FG with anal fistula could simoultaneously treated could help other surgeons around the world dealing with this relevant clinical scenario. 

## 6. Conclusions

In conclusion, the management of FG, mainly if associated with a rectal fistula, needs an aggressive step-up multidisciplinary approach. Surgical debridement and broad-spectrum antibiotics remain the pillars of effective treatment. Hyperbaric Oxygen Therapy in a referral center could speed up the healing process. The presented 3-step eso-endo-VAC new technique could facilitate the simultaneous treatment of open perineal wounds associated with rectal fistulas. Moreover, a lack of evidence regarding the role of NPWT in NSTI invites a deeper analysis of this subject. Randomized controlled trials are necessary to support the NPWT effectiveness.

## Figures and Tables

**Figure 1 jpm-12-01695-f001:**
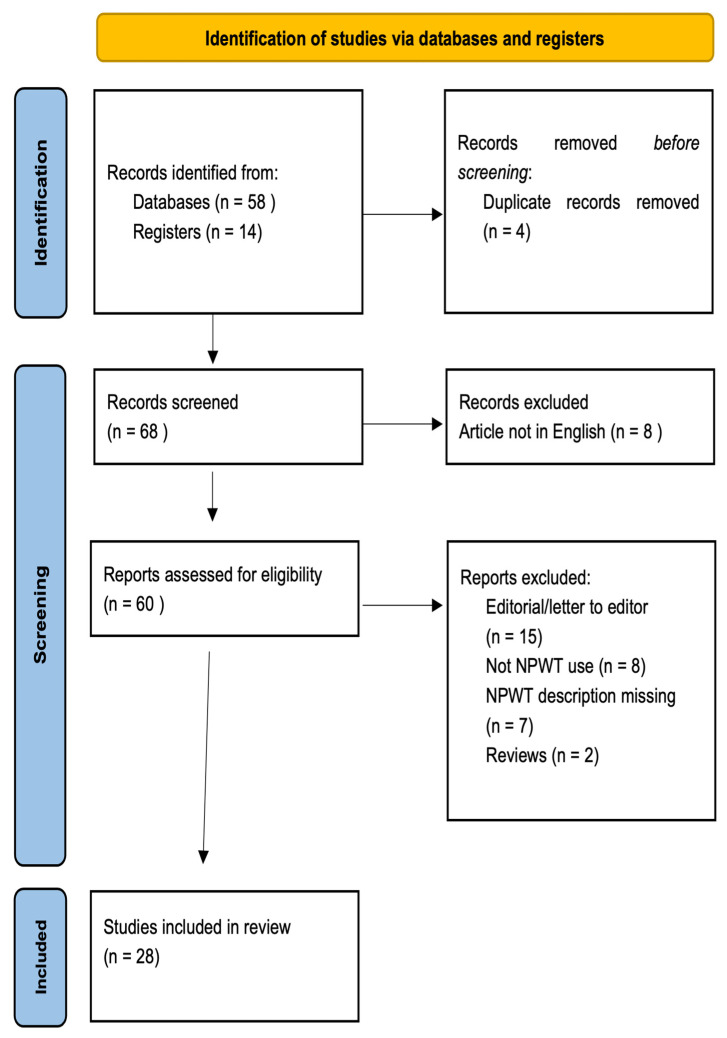
PRISMA flow diagram [12]. For more information, visit: http://www.prisma-statement.org/; date of first access: 10 February 2022.

**Figure 2 jpm-12-01695-f002:**
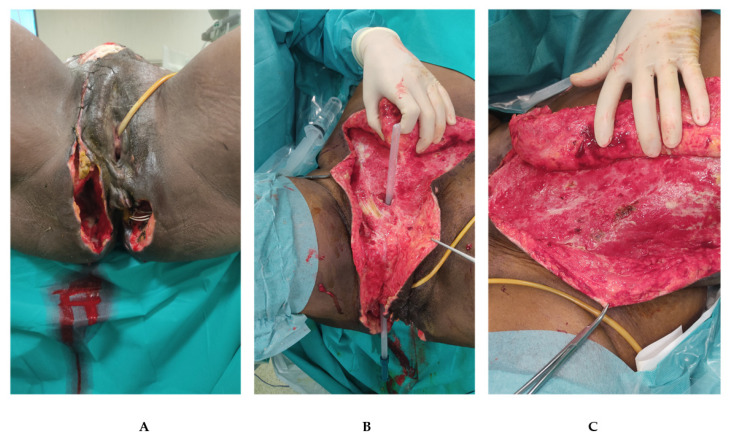
**Patient presentation at the time of surgical debridement**. (**A**): Perineal region presents bilateral cavity with skin and soft tissue necrosis. A vessel loop (with loop) was placed in the rectal fistula to underline the continuity with the left caviy. (**B**): rectal tube placed in the right inguinal canal explaining the reason for the spreading to the abdominal region of the infection. (**C**): Left flank view after the first debridement.

**Figure 3 jpm-12-01695-f003:**
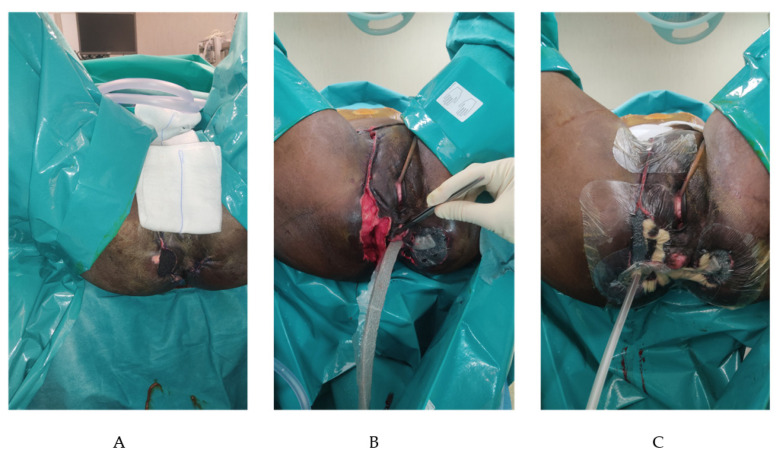
The 3-step eso-endo technique for NPWT positioning in the simultaneous treatment of open perineal wound and rectal fistula. After a complete debridement of necrosis and cleaning of the wound, primary closure of the healthy tissue is performed where feasible. After that, the three-step technique of NPWT placement is achieved. (**A**): A dressing sponge is placed in the cavity in continuity with the rectal fistula (left side) and the other cavity (right side). (**B**): A Steri-Drape fully covered tubular sponge is placed towards the rectal fistula bridging the two previously positioned sponges. So, the suction direction is from the cavity to the rectum, avoiding the solution’ enlargement and promoting the combined healing of both cavity and fistula. (**C**): Steri-drapes are placed, and negative pressure therapy is started. The pressure was stated at −125 mm/hg in a continuous mode and renewed every 48–72 h.

**Figure 4 jpm-12-01695-f004:**
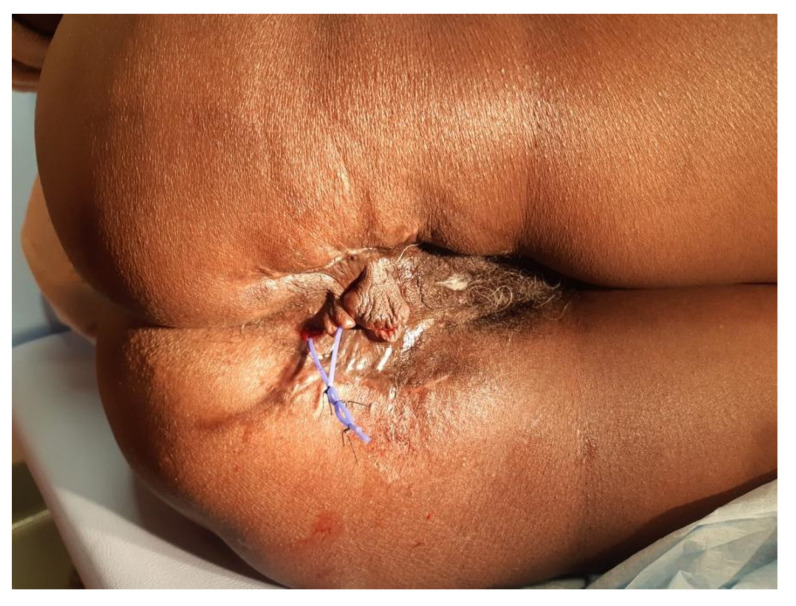
The outcome of the perineal region with the seton placed.

## Data Availability

The data presented in this study are available on request from the corresponding author. The data are not publicly available to preserve confidentiality.

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
