# Peer review of "Negative Pressure Wound Therapy for the Treatment of Fournier’s Gangrene: A Rare Case with Rectal Fistula and Systematic Review of the Literature"

_jpm, 2022, doi:10.3390/jpm12101695_

Round 1
Reviewer 1 Report
Many thanks for the opportunity to review this manuscript.
The authors have presented a case report of a patient with Fournier’s gangrene with rectal fistula, and then attempted to provide a narrative review of cases in the literature, some of which had negative pressure wound therapy applied. All of the included manuscripts are either case reports (single cases), case series (all receiving NPWT) or cohort studies (case series where some patients have NPWT). All included studies had inherent bias and methodological issues so a meta-analysis could not be undertaken. A narrative review is then provided.
If the manuscript is to be published there are several formatting issues that would need to be addressed (too numerous to list here), and sympathetic editing is required of the English grammar.
The title is a little misleading - of the 510 cases included (509 in the literature, and the case presented) only 5 patients (<1%) were described as having a rectal fistula (4 cases in the manuscript by Zhang).
Overall, the authors have been ambitious in trying to provide a coherent review, but they have ultimately failed because of the dearth of good quality data. If the manuscript were to be considered for publication it would need to be significantly restructured to make it easier to comprehend the data presented. It would be easier to read if their case were included before the narrative review. It would be worth considering restructuring Appendix B, and replacing it with a more detailed table of data from the cohort studies to include, in addition to the data already presented, the country of origin, gender mix, presence/absence of risk factors, duration of antibiotic therapy, number of surgical debridements, time to application of NPWT, and the mortality/survival rate. The generic microbiology review (polymicrobial/monomicrobial) adds little. Data from the 10 case series could also be distilled into another table, and a final table with data from the remaining case reports. The narrative review could then be refined.
Some additional thoughts:
Line 43 – the quoted mortality rate should be referenced.
Line 69 – unclear of the use of “spreading”
Line 76 – why was January 2009 chosen as a starting point, and how can the search criteria extend to December 2022?
Line 78 “All reviews of the literature and articles describing NSTIs and FGs treated without NPWT were excluded”.
Line 99 – duration of antimicrobial therapy is not included in the text or in appendix B
Line 101 – NRL score is not defined in the text (assumed to be neutrophil/lymphocyte ratio)
Line 160 – different font size
Results: The narrative review proceeds chronologically from 2009 to 2021, although the publication by Oguz (reference 29) was dated 2015, but comes after publications from 2016, and is dated 2016 in the main text (line 243). Similarly, reference 30 (Chang, FS et al.,) was published in 2018 rather than 2017 (line 247)
Line 161 – The year of publication for Czymek and colleagues is given as 2009 but the reference is from 2010. The cited paper includes 38 patients (not 35) and NPWT was used in 50% (and not 58% as recorded in appendix B). I think the authors are referring to a paper in the American Journal of Surgery (2009) 197; 168-176 where the same authors describe 35 patients (probably the same patients as reference 4) where 19 patients had NPWT (still not 58% as recorded in appendix B). Also, the selection process states that “when more than one article was reported by the same institution and authors, we selected the one with the most extensive series and the most recent” (sic). The authors have clearly done this by including the 2010 reference, but taken data from the 2009 publication, where the details are not included. This should be clarified.
Line 184 – Wagner and colleagues is included as a cohort study but all patients received NPWT, so is this not a case series?
Line 219 – Lee and colleagues did not use shoelaces but SURGI-LOOP applied to both ends of the wound margins in a shoelace manner. Only 3 patients were diagnosed with Fournier’s gangrene
Line 247 – the report by Chang (reference 30) only included those with FGSI scores less than 9 (seen as the cut-off value for severity).
Line 253 – Although Misiakos and colleagues (ref 32) reported on 62 patients with necrotising fasciitis only 29 involved the perineum, and only 4 patients underwent NPWT – unclear which of the patients this included
Line 260 – The author’s initials have been included (unnecessary) but have also been transposed.
Line 264 – Unsure of the use of “topics”
Line 269 – Author’s first name is included in the text.
Line 274 – the publication by Kostovski was published in 2021
Line 305 – suggest “…diagnosis of Fournier’s gangrene was made”
Line 310 – Unusual combination of antimicrobials. Also line 317, why was the antimicrobial therapy not rationalised at that point? Why was it continued for 4 weeks?
Line 338 – suggest “…close the colostomy soon”.
Line 413 and 415 – Author surnames do not need to be in italics
Although NPWT appears to be used with increasing frequency the discussion needs to highlight that there is evidence that NPWT does not improve survival and can significantly increase length of stay. Data on time to wound closure is mixed. The data is inherently biased, and the use of NPWT is often restricted to those with lower initial FGSI scores (less severe infections). Also, as in the case presented, NPWT is often applied after a series of surgical debridement, and not in the acute setting. As highlighted in the conclusion a properly conducted multi-centre randomised control trial of NPWT for NSTI is required. A similar study is required to address the role of HBO.
Figure 1 – the title panels are a little too small for the text. The flow chart states that 32 studies were included in the review, but the text states 28. There are 29 studies in appendix B.
Appendix B is interesting, although it is unclear how the order of the authors has been decided. The author names need to reference number to be included, and it might be easier if the table were as a single table in landscape rather than portrait.
There are several inconsistencies through-out Appendix B:
Page 11: Iacovelli – N/a used instead of N/s
Page 12: In hospital stay (days; NPWT/traditional) – sometimes three values (I assume three individual patients, sometimes a single value (assume single patient with NPWT, but also for case series e.g. Misiakos), sometimes two numbers (median?)
Page 12: Time to closure (days) not clear if from initial surgery, or the application of NPWT. Again, different number of values inputted – same comments as above
Page 13 – FGSI score -Wagner and Chang – two numbers separated by a comma – unclear what this means (median of group etc). Same on page 14 (Hong) and page 15 (Cuccia, Oguz, and Oymaci)
Page 14 Hong – Only 2 of the 20 patients underwent NPWT – should be 10%
Page 14 Tian – Antibiotic use – The initial antibiotic used, and subsequently changed is not documented. The report suggests the patient developed issues with resistance and so antibiotics were discontinued. The cefuroxime (not cefuroxime) was given to cover the STSG, and would no have bene active against at least one the organisms grown from the initial debridement (P.aeruginosa)
Page 14 Perry – Origin of infection – data missing
Page 15 – Perry – HBO – data point missing
Reference 34 (Yucel) – 16/25 had NPWT (64%) not 54% as recorded in Appendix B
Author Response
We want to thank editors and reviewers for their suggestions on our manuscript.
I am attaching our point-to-point letter of response.
Sincerely,
Michele Altomare

Reviewer 2 Report
I would like to congratulate the authors for the detailed description of the clinical case and bibliographic review carried out.
Suggestions:
1) I recommend including “alcoholic liver disease” as an additional risk factor for FG (Introduction and Discussion sections).
2) I recommend adding the LRINEC score in the description of the clinical case, and to indicate whether the patient was hemodynamically stable, and providing basic analytical data.
3) I recommend indicating whether the patient had a rectal tube o Flexi-SealTM system in the period between the placement of the seton (day 1) and the temporal transverse colostomy (day 8).
Author Response

(The authors gave the same response as above.)

Round 2
Reviewer 1 Report
I am grateful to the authors for using my suggestions to improve the manuscript. It now reads better, and the table in appendix B is a significant improvement. I have not gone through the manuscript in fine detail (due to time pressures) although I have spotted a couple of issues.
In the results (1st line) the authors describe 58 eligible articles but the PRISM chart records 68 articles screened and 60 assessed for eligibility.
Appendix B as it is presented is very small! Spelling error in polymicrobial.
These are minor quibbles, but now the manuscript is acceptable for publication
Author Response
We would like to thank R1 for these other really precious suggestions.
We have changed the text according to the revisions.
Please do not hesitate to contact me in case of further necessity.
Sincerely,
Michele Altomare
